# A Study of the Impact of Strategic Human Resource Management on Organizational Resilience

**DOI:** 10.3390/bs12120508

**Published:** 2022-12-13

**Authors:** Jingjing Yu, Lingling Yuan, Guosheng Han, Hui Li, Pengfei Li

**Affiliations:** 1Business School, Shandong University, Weihai 264209, China; 2School of Economics and Management, Harbin Institute of Technology, Weihai 264200, China; 3School of Public Health and Management, Binzhou Medical University, Yantai 264003, China

**Keywords:** strategic human resource management, organizational resilience, self-efficacy, self-management

## Abstract

Organizational resilience is a key capability for modern firms to survive and thrive in the VUCA environment. The purpose of this study is to investigate the mechanism of strategic human resource management on organizational resilience and the mediating and moderating roles of self-efficacy and self-management, respectively, in the relationship between the two. A total of 379 valid questionnaires were obtained from employees of Chinese companies in August 2022, and the data were analyzed using SPSS 22.0 and Amos. The results showed that strategic HRM can effectively contribute to organizational resilience; self-efficacy plays a mediating role in the relationship between strategic HRM and organizational resilience; self-management can effectively contribute to the impact of self-efficacy on organizational resilience; and self-management can hinder the ability of strategic HRM to contribute to organizational resilience. This paper breaks with the previous literature that studied organizational resilience from a single perspective by studying organizational resilience from the perspective of strategic human resource management (SHRM) and verifies that SHRM can be a possible path for Chinese firms to improve organizational resilience.

## 1. Introduction

In recent years, unexpected events such as natural disasters, financial crises, industrial accidents, trade embargoes, and even terrorist attacks have occurred frequently, and sudden “black swans” and “gray rhinoceroses” have led to a business environment that has become more full of volatility, uncertainty, complexity, and ambiguity (VUCA) [1,2]. The still-unresolved novel coronavirus epidemic has caused more than 40% of Chinese companies to lose money or suffer severe losses, and employees to lose their jobs or take pay cuts [3], which has a major negative impact on China’s social and economic development. It also revealed that many Chinese companies are “rigid” but “not resilient”. Organizational resilience is the core capability of today’s enterprises to cope with crises in a volatile, uncertain, complex, and ambiguous (VUCA) market environment. It helps companies to remain sensitive and adaptable to the external environment and to recover and bounce back quickly from the challenging impact of adverse events. Additionally, in the process of reflection and improvement, it goes against the trend to become the key to gain core competitiveness and even steady progress [4,5,6]. Thus, how to enhance organizational resilience in a dynamically changing business environment has become a hot topic for entrepreneurs and scholars to address [7].

A review of the literature related to organizational resilience at home and abroad reveals that the current research on organizational resilience in China is still in its infancy. The research is mainly concerned with the elaboration of the concept and principles of organizational resilience, and the scarce literature on the antecedents of organizational resilience are studied from one aspect, such as management methods, human capital, social capital, business environment, organizational system, etc. [8]. Organizational resilience is a reflection of an enterprise’s comprehensive ability to cope with an uncertain environment, and the study of a single factor cannot comprehensively explain the formation of the mechanism of resilience in an enterprise and cannot well integrate the impact of the interaction of various factors on organizational resilience, which lacks operability in enterprise practice and cannot effectively help enterprises to improve organizational resilience [9]. At present, there is an urgent need to study the formation mechanism of organizational resilience from a holistic perspective considering the comprehensive effect of each antecedent factor and provide a theoretical basis for Chinese enterprises to improve organizational resilience. Strategic human resource management is a system, process, or measure consisting of a series of temporal activities taken in order to fit with the organization’s strategy and long-term development goals and thus maintain competitive advantage. Lengnick-Hall [10] believes that organizational resilience works through the knowledge, skills, abilities, and other attributes of people within an organization, and that strategic human resource management can be achieved by changing management styles, processes, practices, and HR policies, etc. to develop these qualities in employees to enhance organizational resilience.

In exploring the role of modeling the relationship between strategic human resource management and organizational resilience, this paper found the “key” of self-efficacy, drawing on the literature on emotional competence. All HR policies and plans in an organization require employees to take specific actions to achieve the desired goals. Thus, self-efficacy, an essential emotional competency on the part of employees, is critical to the effectiveness of plan implementation. Early warning capability, flexibility during a crisis, and learning and growth capability following a crisis are three core competencies included in organizational resilience [11]. Self-efficacy refers to a strong belief in one’s own ability to comply with a corporate strategy, which enables employees to relieve psychological pressure in time, stabilize the psychological reactions of personnel to adversity, actively obtain environmental resources and external support to optimize human resource allocation, and integrate human resource management practices with corporate strategy organically by focusing on internal personnel selection and appointment, performance evaluation and assessment, and external active recruitment; this approach allows organizations to shape a corporate culture of overcoming difficulties, create dynamic and flexible adaptation mechanisms, construct internal knowledge structures, and actively seek foreign environmental resources, thereby enhancing corporate resilience capabilities by providing solutions to crises and addressing corporate structural problems [12]. Thus, self-efficacy is an essential perspective for the study of organizational resilience that can explain why strategic HRM are able to influence organizational resilience capabilities; therefore, this study intends to discuss the relationship between strategic HRM and organizational resilience as mediated by self-efficacy.

In addition, we cannot ignore the key issue that strategic HRM must be implemented and accomplished through corporate employees no matter what policies are formulated, and corporate organizational resilience capabilities must also function through corporate employees as mediators. Thus, studying the mechanism of strategic HRM’s effect on organizational resilience is inevitably influenced by employees’ work style. As a result of the rapid development of the economy and excellent material abundance of China, employees’ sense of autonomy and their self-working ability are becoming increasingly prominent. According to the traditional HRM model, it is difficult for leaders to supervise and constrain employees. It is more practical to study the effects of strategic HRM with respect to enhancing organizational adaptability and flexibility to the environment from the perspective of employees’ sense of autonomous work. Therefore, this paper uses self-management as a moderating variable to determine whether self-management plays a moderating role in the relationship between strategic HRM and organizational resilience.

According to a report by Fortune magazine, the average life expectancy of small enterprises in China is 2.5 years, and the average life expectancy of large enterprises is 7-8 years, which is far behind those of European and American countries. From the side, it shows that Chinese enterprises lack the ability of resilience to cope with the crisis. Especially under the impact of the novel coronavirus epidemic, many enterprises have experienced serious losses or even bankruptcy, resulting in employee pay cuts and unemployment, adding a heavy burden to China’s social stability and economic development. In summary, this paper takes conservation of resources theory and self-cognitive theory as the theoretical basis and empirically investigates the intrinsic mechanism of strategic human resource management on organizational resilience. It provides references for Chinese companies to enhance organizational resilience.

The main contributions of this study include two main aspects. On the one hand, this study can enrich the literature related to the study of strategic human resource management and organizational resilience; on the other hand, the research results of this paper can guide the managers of Chinese enterprises to formulate strategic human resource planning, coordinate all resources of human, financial and material resources, optimize enterprise processes, improve enterprise management policies, increase enterprise innovation, etc., so as to enhance organizational resilience.

## 2. Literature Review and Research Hypothesis

### 2.1. Strategic Human Resource Management and Organizational Resilience

Strategic HRM was developed to facilitate the strategic management of organizations, and Wright and McMahan [13] give a more representative definition of this concept. They consider strategic HRM to represent an organization’s plans for human resource deployment and behavioral norms with the aim of achieving the organization’s goals. This definition emphasizes both the vertical and horizontal fit of strategic HRM: vertically, strategic HRM refers to the match between and mutual adaptation of HRM practices and the organization’s strategic management process, whereas horizontally, strategic HRM emphasizes the coherence among various HRM practices based on the planned action model. The vertical and horizontal fit of strategic HRM ensures that HRM is fully integrated into strategic planning to guarantee that HR policies and practices are generally accepted and widely used by managers and employees, such that companies can obtain inimitable or alternative competitive advantages by leveraging their HR strengths [9,14]. According to conservation of resources theory, strategic HRM, as a strategic organizational resource, represents an organic combination the talent resource elements in an organization as well as the allocation of resources among members of the organization; thus, strategic HRM emphasizes the flexible adjustment of staffing policies and practices, training and development programs, performance standards, selection criteria, and rewards and punishments in response to changes in external contexts, thus providing strategic tools to promote resource integration, crisis prevention and control, and learning and innovation in organizations [15].

The concept of resilience originated in the fields of physics, ecology, and environmental science, and Meyer [16] first introduced this concept into the field of management, thereby opening up a new chapter in the study of organizational resilience, which was quickly and widely studied in the fields of crisis management, disaster management, and high-reliability organizations. Previous research on organizational resilience has been focused on two main research perspectives: the rebound perspective and the rebound + overtake perspective [10]. The rebound perspective views organizational resilience simply as the ability of the organization to recover from an accident, stress, or crisis to return its original state, i.e., the ability of an organization to take countermeasures to return to its precrisis level of performance. The rebound + overtake perspective views organizational resilience not merely as the organization’s ability to respond to challenges and changes to return to its original state but also as the organization’s ability to develop new capabilities or create new opportunities for the organization to continue to thrive and grow [17]. This paper considers organizational resilience to be a dynamic and flexible organizational capacity that allows organizations to survive, adapt, recover, and even return to prosperity in an adverse environment. Lengnick [18] claimed that organizational resilience capacity is rooted in the psychology and behavior of individual employees. Employees’ knowledge, skills, and strengths regarding their values, mindsets, levels of stress tolerance, and innovation abilities are essential sources of organizational resilience [18,19]. These employee qualities and capabilities are closely related to the individual’s ability to adapt to dynamic environments and to develop creative solutions to resolve crises. Employee resilience is an important source of and foundation for organizational resilience; thus, organizations can enhance their organizational resilience capabilities by developing employee resilience.

In this paper, we argue that strategic HRM can influence individual resilience and thus enhance organizational resilience via the development of HRM policies and practices that match both the external environment and organizational goals [20]. Specifically, the effects of HRM policies and practices on organizational resilience can be elaborated in terms of three aspects of human resources: human capital, social capital, and psychological capital [15].

First, human capital primarily includes the physical quality and physical health of employees on the one hand and the knowledge, skills, and experience possessed by employees on the other hand. In a crisis, members of an organization can make timely judgments and actions based on the knowledge, skills, and experience they possess to change the organization’s passive situation as much as possible, thus influencing its resilience [19]. The exchange of knowledge and experience among organizational members and their interactions can promote the formation of the collective cognition of the organization. This collective cognitive ability encourages organizational members to cooperate tacitly, trust each other, and unite in the face of adversity, thus developing the unique organizational ability of the enterprise to cope with crises and affecting the organizational resilience ability [21].

Second, social capital is a potential resource possessed by the organization within the social network system, which is essentially an environmental factor that is mainly divided into internal environmental factors (e.g., colleague relationships, learning atmosphere, team spirit) and external environmental factors (e.g., partnership with suppliers or distributors, flexible external information system) [22]. Social capital can increase the levels of coordination and cooperation that employees exhibit in their work, which in turn increase the motivation and efficiency of the organization with respect to coping with a crisis. Moreover, social capital can be used to obtain resources and information from the external environment that are critical for crisis resolution and the reallocation of resources both inside and outside the organization, thus enhancing organizational resilience and mitigating the negative impacts of the crisis for the organization.

Finally, employees with high psychological capital can withstand the tremendous pressure entailed by a crisis and face challenges and changes with a positive and confident attitude, create a good organizational climate, and apply their knowledge and skills based on the local conditions to create opportunities for the organization to survive and grow in the face of adversity, which has a significant impact on the organization’s ability to enhance its resilience and obtain competitive advantage [23]. In addition, previous studies have demonstrated that strategic HRM that is well matched with the organizational environment, strategic planning, and corporate culture is closely related to organizational resilience. For example, Shafer et al. [23] found that when organizational HR practices are aligned with organizational values, organizations can promote organizational agility through staffing policies, personnel training, career development programs, and performance standards, thereby enhancing organizational resilience. Okuwa [24] found positive relationships among training, human resource development, and organizational resilience. Mienipre [25] found that talent management was significantly correlated with organizational risk monitoring and crisis response capacity. In summary, the following hypothesis is proposed.

**Hypothesis 1** **(H1).**
*Strategic human resource management has a positive effect on organizational resilience.*


### 2.2. The Mediating Role of Self-Efficacy

The concept of self-efficacy was introduced by Bandura [26], an American psychologist who believed that self-efficacy represents an individual’s subjective evaluation and perception of his or her abilities, which in turn influences the individual’s behavioral choices, beliefs regarding success, and level of effort, and can to some extent determine the individual’s ability to fulfill the requirements of a particular job; that is, self-efficacy is dynamic and can change due to different levels of access to external resources, the acquisition of new knowledge and skills, or an increase in experience. According to previous studies, the factors affecting self-efficacy mainly include the following. (1) Individuals’ ability levels are evaluated prior to performing certain activities; individuals evaluate their own ability based on their past successes or failures, such that individuals who exhibit a strong sense of self-efficacy do not deny their ability due to occasional failures but rather search for the causes of environmental factors, strategies, and experiences and adjust their future actions accordingly. (2) Individuals who observe the behavior of others and encounter people with similar abilities who have achieved success can greatly enhance their own self-efficacy and increase their firm belief in achieving success. (3) Individuals receive evaluations, encouragement, and self-motivation from others. Evaluations or encouragement based on the facts of the situation can increase the individual’s belief in his or her ability to accomplish the goal. (4) The individual’s own emotional and physiological state also affects self-efficacy, such as the ability to remain calm under tremendous pressure, avoid exhibiting arrogance, analyze the pros and cons of the actual situation, and make the most appropriate decision, which can increase the individual’s ability to accomplish the goal as well as his or her sense of self-efficacy [26].

Self-efficacy is an important component of human capabilities that can influence individuals’ perceptions, ways of thinking, motivation, and actions [27]; in addition, it varies with people’s knowledge and external environment [28]. Thus, organizations can improve employees’ self-efficacy by implementing human resource practices such as communication, training, sharing successful experiences, and providing opportunities for success. For example, organizations can increase employees’ relevant work experience by providing training and organizational learning [29]. When employees are trained in job-related practices, they are able to acquire relevant job knowledge and information that can enhance their self-efficacy to perform their jobs competently. Second, employees’ self-efficacy can be stimulated by sharing the successful experiences of colleagues with similar abilities to enhance their beliefs in their ability to overcome specific job difficulties and their efforts to do so, thus moderating the empowerment of employees and providing them with opportunities to grow and succeed to ensure that employees feel supported by the organization and trusted by their leaders; this approach increases employees’ sense of organizational belonging and self-efficacy, thus allowing the organization to take full advantage of employees’ knowledge and skills and to face challenges and cope with stress actively. Strategic human resource management refers to the alignment of organizational strategic planning with human resources, which is used to guide human resource practice activities and is frequently considered to be an essential factor influencing the cognitive, motivational, and affective processes of self-efficacy [30]. Organizations can ensure sound planning and develop action plans for future operations by engaging in HR activities such as training, sharing successful experiences, role models or motivation, and developing employees’ confidence in dealing with dynamic environmental challenges and complex work. In summary, the following hypothesis is proposed.

**Hypothesis 2** **(H2).**
*Strategic HRM has a positive effect on self-efficacy.*


According to conservation of resources theory, self-efficacy, as an essential psychological resource, is closely related to employees’ self-beliefs and can motivate them to accept challenges and persevere in the task of accomplishing their work goals [31]. Thus, when facing complex tasks, on the one hand, self-efficacy can strengthen employees’ determination and confidence to complete tasks and allow them to unite their colleagues actively, integrate relevant resources and information, and courageously face difficulties and challenges [32]; on the other hand, self-efficacy can motivate employees to self-regulate in a timely manner, relieve tension and anxiety, and reallocate resources and set goals based on the specific situational conditions at hand to ensure that difficulties can be broken down into simple goals and achievable work objectives [33,34,35]. In addition, employees who exhibit a high sense of self-efficacy are skilled at using new methods and ideas to solve unconventional problems, thus enabling the organization to find alternative ways of surviving situations of adversity and contributing to the organization’s resilience [35]. In conclusion, self-efficacy enables employees to believe in their ability to work in situations of adversity, recover quickly from anxiety, and invest the necessary effort and creativity to accomplish challenging tasks. Therefore, the following hypotheses are proposed.

**Hypothesis 3** **(H3).**
*Self-efficacy has a positive effect on organizational resilience.*


**Hypothesis 4** **(H4).**
*Self-efficacy plays a mediating role in the relationship between strategic human resource management and organizational resilience.*


### 2.3. The Moderating Role of Self-Management

According to self-cognitive theory, individuals have certain values, beliefs, knowledge systems, and behavioral norms. Individuals form their unique control systems based on these internal resources and accordingly set goals, engage in self-assessment, and exhibit self-motivation as a means of guiding their work activities, i.e., self-management [36]. The awareness of the practice community that organizational control and supervision must be achieved by influencing the self-management system to achieve this goal, i.e., by harmonizing organizational control and individual motivational orientation, is increasing [37]. Self-management refers to the process by which employees set goals, take positive actions, and engage in a series of behaviors, including self-monitoring and evaluation as well as self-reward and punishment, to promote their own intrinsic self-worth based on their personal needs. Self-management results from the interaction of individual cognition, behavior, and the external environment. Bandura [38], in developing social cognitive theory, proposed that individuals exhibit self-rationality, that is, that the individual’s response to the external world is not mechanical and passive but rather represents a form of goal-oriented behavior following self-regulation of and self-reflection on their activities; in addition, the achievement of such a goal can allow the individual to obtain self-worth and meaning (such as monetary or spiritual rewards, social needs, or self-actualization). Bandura’s model of individual self-management [39] includes three components: self-observation, self-assessment, and self-response. The process of self-observation involves actively identifying the quality, quantity, and frequency of the performance accomplishment of other individuals and comparing those individuals with oneself to make an objective assessment of one’s own work ability; the process of self-assessment entails comparing one’s actual performance with the company’s performance standards, thereby assessing one’s own performance and developing strategies for improvement; and the process of self-reaction implies rewarding and punishing oneself according to the results of the assessment as well as reflecting on and improving oneself continuously.

According to conservation of resources theory, organizational resilience is an essential intangible resource that allows the organization to survive and develop in adverse situations, thus enabling organizations to make decisive decisions in dynamic situations, flexibly deploy their internal and external resources, and take appropriate actions to ensure that the organization is always able to adapt to the business environment and obtain competitive advantage [11]. In times of crisis, only if the organization is united, determined, and confident can it seize the fleeting moment, make decisive decisions, and act efficiently to take full advantage of its own resilience. A high level of self-management ability on the part of employees, with good self-cognitive ability and the ability to obtain and process environmental information to ensure that they can quickly judge the situation in times of crisis and work in an orderly manner based on the situation, is target management. Thus, a high level of self-management can increase employees’ sense of psychological security and self-efficacy in times of crisis, thereby enhancing the resilience of the organization. Specifically, on the one hand, the process of self-management is driven by employees’ intrinsic values, and the achievement of the organization’s goals is a testament to employees’ self-worth [40]. Employees view the challenges presented by adversity as opportunities to prove their own ability and value. They view work as their responsibility and believe in accomplishing challenging goals by taking full advantage of their professional skills and creativity and working with other members of the organization to deal with environmental challenges and smoothly survive crises, thereby enhancing organizational resilience. On the other hand, employees with high levels of self-management are skilled at assessing their own abilities and performance levels or those of others as well as setting reasonable work goals and developing reasonable action strategies based on the resources and information that they obtain because employees who are skilled at self-management tend to take the initiative to collect and process environmental information, remain sensitive to the external environment and organizational operations, and ensure that they are always needed by the organization as a means of maintaining their competitive positions in the organization. In times of crisis, when the organizational landscape changes, employees quickly orient themselves to their goals with self-management ability, integrate their accessible resources, develop reasonable action plans and smoothly execute them, reduce their confusion and anxiety, increase their self-efficacy, and take full advantage of organizational resilience [41]. In conclusion, self-management can enhance the contribution of self-efficacy to organizational resilience. In summary, the following hypothesis is proposed.

**Hypothesis 5** **(H5).**
*Self-management plays a positive role in regulating the impact of self-efficacy on organizational resilience.*


Previous studies have shown that the application of self-management in organizations can reduce the costs of business supervision and management and improve business performance and employee well-being, among other effects [42]. However, self-management is not effective in all situations [43]. Employee self-management is based on mutual trust between leaders and employees, such that leaders trust employees to be capable of accomplishing the established goals, and employees trust that they will receive set rewards for accomplishing such goals [44]. However, critical moments that require companies to demonstrate their resilience to cope with difficult times can lead to changes in companies’ human resource management plans and thus in their goals and development direction as well as the reshuffling of personnel rights and interests within such companies. In this situation, employee self-management hinders the ability of strategic human resource management to promote the company’s organizational resilience capabilities. Specifically, first, according to conservation of resources theory, the achievement of self-management goals requires the input of individual and organizational resources [45]. Crises cause the achievement of goals to be rife with uncertainty. Employees have negative attitudes toward the implementation of the company’s strategic human resource plan and corporate goals due to their desire to prevent their resources from being lost. In addition, organizational resources become scarcer and more difficult to acquire in a crisis. Employees tend to compete for internal resources to maintain their existing resources and rights, which strains the relationships among people within the organization and is not conducive to communication, cooperation, and knowledge sharing among members of the organization. In contrast, organizational resilience requires a high degree of team cohesion, mutual trust, assistance, and cooperation and thus is not conducive to organizational resilience. Second, the adjustment of strategic HR policies in times of crisis can lead to the reformulation of individual goal management. Employees’ internal self-actualization and self-growth are essential drivers of self-management goals. Once corporate goals deviate from individual goals, employees’ actions may impede or even prevent the implementation of strategic HRM plans, thus rendering the organization unable to deploy people and resources rapidly and perhaps even causing the organization to miss the best time to act, which is not conducive to the development of organizational resilience [46]. Finally, strategic HRM is a management approach that aligns HRM with corporate strategy. Corporate strategy often takes the form of management involving multiple goals, such as corporate performance, social responsibility, and brand image. The complexity of work and teamwork cause corporate goals to become indistinguishable or unclear, which makes it difficult for employees to set and implement their personal self-management goals, thereby causing them to become confused and uncomfortable and to experience self-doubt or negative emotions, which is not conducive to the development of organizational resilience. In summary, the following hypothesis is proposed.

**Hypothesis 6** **(H6).**
*Self-management negatively affects the impact of strategic human resource management on organizational resilience.*


In summary, based on conservation of resources theory and self-cognitive theory, this paper constructs a moderated mediation model, as shown in Figure 1, and examines the relationships among strategic human resource management, self-efficacy and self-management, and organizational resilience.

## 3. Methodology

### 3.1. Study Sample

This study was conducted to investigate the organizational resilience of enterprises within China. In order to guarantee the accuracy, reliability of data, and wide distribution of the research sample, this study follows the principle of randomness to select the employees of enterprises with different industries, ages, education levels, positions, and income statuses in several cities within China. Due to the special national conditions of China, there are major disparities in the development levels of various regions, so the sample source of this paper includes developed large cities, such as Beijing, Shanghai, Shenzhen, Guangzhou, etc., medium development level cities, such as Jinan, Qingdao, Dongguan, Huizhou, Haikou, Lanzhou, etc., and developing small cities, such as Jiuquan, Weihai, Hami, etc. For the convenience of sample collection, a combination of online and on-site distribution was chosen for this study. In order to guarantee the authenticity and accuracy of the acquired data, a partial reverse setting of the question items was used. It was filled out voluntarily and anonymously to reduce the concerns of those who filled it out. A lottery link was also included with the questionnaire to incentivize the completion of the questionnaire. A total of 441 questionnaires were collected for one month starting from August 2022, excluding invalid questionnaires that were completed too quickly, filled out incorrectly, had omitted answers, or were duplicates. 379 valid questionnaires were obtained, for a return rate of 86%. The gender distribution of the sample was 53% males and 47% females; the age distribution included 16.9% of participants aged 25 and below, 37.2% aged 26–35, 28.5% aged 36–45, 14.8% aged 46–55, and 2.6% aged 55 and above; the education distribution included 20.6% of participants with a high school/junior college education, 43.3% with bachelor’s degrees, 10.3% with master’s degrees, and 2.1% with doctoral degrees.

### 3.2. Variable Measurement

All variables included in this study questionnaire were measured using the seven-point scale developed by Richter. All the scales used in this paper are well-established scales with good reliability and validity that have been validated many times in the Chinese cultural context. Additionally, a small sample of 73 people was taken for pre-study. Afterwards, two professors and three PhD and MSc students in the field of business management and human resource management examined and adjusted the new questionnaire according to the research questions, validity, and reliability of the questionnaire results, Chinese cultural background, and readability.

Organizational resilience: this variable was measured using a 15-item organizational resilience scale developed by Xiu’e Zhang et al. [47] in the context of China. This scale contains items such as “ability to adapt and creatively solve problems when a crisis occurs” and “ability to access needed resources quickly to address challenges in times of crisis”.

Strategic human resource management: this variable is assessed using a 19-item scale based on Delery’s Strategic Human Resource Management Scale [48], adapted to the Chinese cultural context, which contains items such as “Individuals in this job have clear career paths within the organization“ and “Individuals in this job have very little future within this organization (reverse-coded)“.

Self-management: this variable is assessed using a 10-item scale based on Renn ‘s self-management scale [49], adapted to the Chinese cultural context, including “I set specific goals for myself at work”, “I establish challenging goals for myself at work”, and “I clearly define goals for myself at work”.

Self-efficacy: this variable was measured using an eight-item scale developed by Chen et al. [50]. This scale contains items such as “I will be able to achieve most of the goals that I have set for myself” and “When facing difficult tasks, I am certain that I will accomplish them”.

Control variables: this paper investigates the effects of organizational human resource management policies, self-efficacy, and self-management on organizational resilience from the perspective of human resource management. To make the questionnaire data as accurate as possible, the gender, age, and education of employees are used as control variables in this paper to reduce the influence of errors on the analysis of the relationships among variables.

## 4. Results

### 4.1. Common Method Biases Test

This study draws on Podsakoff et al. [51] to procedurally reduce homogeneous method bias by selecting different spatial survey respondents, anonymous surveys, and partial question item reversal settings at the time of data acquisition. Harman one-way analysis of variance was used to measure the presence of severe common method bias. The results of the SPSS 22.0 test revealed a total of six factors with eigenvalues greater than one for the unrotated exploratory factor analysis. Additionally, the maximum factor variance explained was 28.22%, which was much less than 40%; thus, there was no serious common method bias in this study.

### 4.2. Reliability and Validity Tests

First, this study used SPSS 22.0 and AMOS statistical software to analyze the data from 379 samples. Internal consistency tests were conducted based on the criteria of whether the coefficient of internal consistency was greater than 0.7 and whether the coefficient of internal consistency would increase after the deletion of a question item. The test results are described in Table 1. The Cronbach’s α values of all variables are above 0.90, and the deletion of any question item does not increase the Cronbach’s α value significantly, indicating that the variables have good internal consistency. The CR values are all greater than 0.90, and the AVE values are all greater than 0.55. This indicates that the variables have good composite reliability.

Second, this study developed confirmatory factor analysis models for strategic human resource management, self-management, self-efficacy, and organizational resilience and conducted confirmatory factor analysis on the research models using AMOS. The results showed that all model indicators met the statistical benchmark values (χ^2^/df = 2.658, RMSEA = 0.066, CFI = 0.905, IFI = 0.905), thus indicating that the model goodness of fit well. In addition, the fit indices of the randomly selected two-factor model and those of one-factor and three-factor models were compared, as shown in Table 2. The results showed that the fit indices of the original model were significantly better than those of the one-factor, two-factor, and three-factor models, thus indicating that the original model had good discriminant validity.

### 4.3. Descriptive Statistics and Correlation Analysis

The mean and standard deviation of each variable as well as the correlations among all the variables were analyzed using SPSS 22.0, and the results of this analysis are shown in Table 3. There was a positive and strong correlation between strategic HRM on the one hand and organizational resilience (r = 0.722, *p* < 0.01) and self-efficacy on the other (r = 0.676, *p* < 0.01); the relationship between self-efficacy and organizational resilience (r = 0.711, *p* < 0.01) also exhibited a positive and robust correlation, thereby providing preliminary evidence to support the research hypotheses.

### 4.4. Test of Mediation Model with Moderation

In this paper, we refer to Wen [52] with the moderated mediation model test method to test the mediation model first, and, on the basis of significant mediation effect, we conduct the moderated mediation model significance test to verify whether each model proposed in this paper is significant.

First, this study tested the mediating effect on the relationship between self-efficacy on strategic HRM and organizational resilience using Model 4 (mediating model) in the SPSS macro developed by Hayes [53]. The results of this test are shown in Table 4. Strategic HRM has a significant positive effect on organizational resilience (B = 0.711, t = 19.952, *p* < 0.001); strategic HRM has a significant positive effect on self-efficacy (B = 0.568, t = 17.180, *p* < 0.001); and self-efficacy has a significant positive effect on organizational resilience (B = 0.459, t = 9.098, *p* < 0.001). In addition, the upper and lower limits of the bootstrap 95% confidence intervals pertaining to the direct effect of strategic HRM on organizational resilience and the mediating effect of self-efficacy do not contain 0, as shown in Table 5, thus indicating that strategic HRM affects organizational resilience not only directly but also indirectly via the mediating effect of self-efficacy, with the direct and indirect effects accounting for 63% and 37% of the total utility, respectively.

Second, the moderated mediation model was tested using Model 15 in the SPSS macro prepared by Hayes (2012) [53]. The results of the test are shown in Table 6 and Table 7. After including self-management in the model, the product term of strategic HRM and self-management has a negative effect on organizational resilience (B = −0.144, t = 6.617, *p* = 0.01). Furthermore, the moderating effect of self-management contains 0 between the upper and lower limits of the bootstrap 95% confidence intervals at the eff1 (M − 1SD) level. In comparison, this effect does not contain 0 between the upper and lower limits of the bootstrap 95% confidence intervals at the eff1 (M + 1SD) level, thus indicating the significant moderating effect of self-management. The product term of self-efficacy and self-management positively affected organizational resilience (B = 0.137, t = 6.617, *p* = 0.001). Further simple slope analysis indicated that the effect of strategic HRM on organizational resilience tends to decrease gradually as the level of self-management increases and that the effect of self-efficacy on organizational resilience tends to increase in this context, as shown in Figure 2a,b.

## 5. Discussion

Based on the conservation of resources theory and self-cognitive theory, this study takes employees in Chinese culture as the research object and explores the mechanism and boundary conditions of strategic human resource management on organizational resilience. The three aspects of human capital, social capital, and psychological capital are explained to ensure that the human resources of a company fit with the corporate strategy to ensure that the strategic goals of the company match with the external environment, and that the internal resources are rationally allocated to promote the organizational resilience. Self-efficacy, as an emotional ability, is an employee’s attitude and belief about the company’s ability to cope with crises. Organizational resilience is a corporate soft capability embedded in employees’ knowledge, skills, and traits. Thus, employees’ beliefs about achieving strategic human resource management goals will influence employees’ performance in times of crisis and thus the ability to perform with organizational resilience. Therefore, the potential impact of self-efficacy on the performance of organizational resilience capabilities cannot be ignored. The impact of self-management on organizational resilience is uncertain. Self-management can enhance the positive impact of strategic HRM on organizational resilience but hinders the positive impact of self-efficacy on organizational resilience.

(1)Hypothesis 1, that strategic HRM in the Chinese context facilitates organizational resilience, was confirmed. Facing a VUCA environment, business operations are fraught with many uncertainties, a point which is especially salient to this study since it was conducted in the middle of the novel coronavirus pandemic, which has had a massive impact on the global economy and people’s lives. To address this major crisis that can reshape the global economic landscape, it is imperative for companies to adjust their corporate strategies and long-term development plans, encourage their employees to respond to the associated challenges actively, and transform the crisis into an opportunity for growth. The empirical study of strategic human resource management and organizational resilience in the face of crisis shows that strategic human resource management can actively transform corporate development strategies, reorganize and reallocate corporate human resources, lead companies to adapt to changes quickly, act flexibly and innovate actively, and have a positive effect on the improvement of organizational resilience. Accordingly, strategic human resource management is an effective way in which enterprises can ensure their survival and obtain competitive advantages in the face of a crisis.(2)This study tested hypothesis 2, that strategic human resource management has a positive effect on self-efficacy, hypothesis 3, that self-efficacy has a positive effect on organizational resilience, and hypothesis 4, that self-efficacy mediates the effect of strategic human resource management on organizational resilience. Based on the argument that strategic human resource management positively affects organizational resilience, this study further argues that strategic human resource management can enhance organizational resilience by increasing employees’ self-efficacy. Self-efficacy refers to an employee’s strong belief in his or her own ability to do his or her job and accomplish the associated tasks. Self-efficacy enables employees to act rationally in times of crisis, to believe that the company has the strength to deal with the crisis, to respond positively to the company’s HR policies and practices, to unite with colleagues, and to dare to solve corporate problems in innovative ways. Self-efficacy enables the company’s strategic human resource management policies and practices to be implemented quickly throughout the company, thereby enhancing the company’s operations and flexibility in times of crisis and enabling the organization to recover quickly from a crisis and respond to a variety of environmental challenges, thus enhancing the organizational resilience that allows the organization to deal with complex environments.(3)Hypothesis 5 was tested, that is, the positive moderating role of self-management in the effect of self-efficacy on organizational resilience. Self-management has a nonnegligible impact on the effect of self-efficacy on organizational resilience. The achievement of corporate strategic goals is ultimately based the actions taken by employees at work, and the self-management ability of employees is related to the efficiency and effectiveness of policy implementation. Employee self-management motivates employees to combine corporate goals with their own internal needs, set their own goals, actively access and use external information and resources, assess the gaps between goals and actual performance as well as the difficulties associated with crossing those gaps, and choose creative action paths to achieve their goals. Thus, self-management ability can enhance the organization’s sensitivity to the external environment, thus allowing the organization to prepare for crises in advance to ensure that employees can act with plans and goals in times of crisis, thereby enhancing their self-efficacy and making full use of their creativity and professional skills; given such preparation, the organization can smoothly survive the crisis and continue normal operations or even increase the prosperity of the enterprise.(4)Hypothesis 6 was also tested, that is, the negative moderating role of self-management in the effect of strategic HRM on organizational resilience. Self-management negatively influences the impact of strategic human resources on organizational resilience. Previous research on self-management has focused on the positive effects of self-management on business management, such as its effects on business performance, employee satisfaction, employee happiness, and creativity. However, this study finds that employee self-management capabilities at the strategic level may be detrimental to organizational resilience. The original driving force behind the role of self-management is rooted in the deep-seated needs of employees. In times of crisis, if adjustments to corporate strategies and resource reorganizations deviate from the goal of self-management, employee self-management may impede or jeopardize the implementation and achievement of corporate strategic goals. Self-management causes the organization to become slow to act, rigid in its operations, and inflexible and insensitive in times of crisis, and it is detrimental to the development of organizational resilience.

Based on these findings, this paper argues that strategic human resource management is conducive to the enhancement of organizational resilience and is a possible way in which organizations can cope with potential crises and turbulent business environments. Strategic HRM allows companies to create innovations in their organizational staffing structures and systems actively, thereby enhancing the ability to self-repair and self-rebound at the organizational level; it allows companies to respond to the diverse and constantly changing needs of the market and customers and enhance the adaptability and flexibility of the organization, which is crucial for the organization’s competitiveness in the market.

## 6. Conclusions

Strategic human resource management facilitates organizational resilience capacity enhancement and is a possible path for organizations to respond to potential crises and turbulent business environments. Strategic HRM facilitates companies to actively innovate their organizational staff structure and system, enhance the ability to repair and rebound at the organizational level, respond to the diversified and changing needs of the market and customers, and enhance the adaptability and flexibility of the organization to the market. This is the reason why many companies are consciously implementing strategic human resource management. Thus, strategic HRM is a possible path for Chinese companies to enhance organizational resilience.

### 6.1. Theoretical Contributions

(1)This paper expands the conservation of resources theory and discusses important antecedent variables that facilitate the organization’s ability to exhibit organizational resilience. Organizational resilience is an essential resource and capability that allows companies to adapt to changes actively following a crisis, seek opportunities for survival and innovation, and overcome difficulties and achieve counter prosperity. In a dynamic and changing business environment and given human-centered management trends, it is crucial to clarify the manner in which strategic human resource management can enhance organizational resilience. Managing and utilizing the company’s employees well in a manner that takes advantage of the company’s talent and allows the company to cope with an unpredictable business environment has become a hot topic for both corporate managers and academic researchers. This paper focuses on the ways in which a human resource management model that fits with corporate strategy can enhance employees’ self-efficacy and thus organizational resilience, thereby providing a new perspective on the relationship between strategic human resource management and organizational resilience, theoretically considering possible ways of enhancing organizational resilience, and helping expand research on the mechanisms underlying the impact of strategic human resources.(2)This paper validates the important influence effect of self-efficacy, and it explores the relationship between strategic human resource management, self-efficacy, and organizational resilience from the perspective of conservation of resources theory and self-cognitive theory, using strategic human resource management as an antecedent variable of self-efficacy, which helps to understand the intrinsic correlation between strategic human resource management, self-efficacy, and organizational resilience in depth. The mechanisms of how strategic HRM affects organizational resilience have been unclear in past previous research. This paper explores the “black box” of the relationship between the mechanisms of strategic HRM’s impacts on organizational resilience through the self-efficacy variable and highlights the vital role and value of self-efficacy in organizational resilience.(3)This paper analyzes the theoretical mechanisms and boundary conditions according to which organizational resilience can function in crises. Regardless of the uniqueness and effectiveness of the strategies and responses that are adopted by enterprises, these strategies and responses must be implemented and facilitated by employees. Therefore, in times of crisis, enterprises should pay more attention to employees’ psychology, attitudes, and abilities, stimulate their creativity and motivation, and take the best path of action. Therefore, this paper includes self-management as a moderating variable to deepen our understanding of organizational resilience at the enterprise human resource management level. Through theoretical extrapolation and empirical research, the paper reveals that employees’ self-management is not conducive to the promotional effect of strategic HRM on organizational resilience, a conclusion which differs from the findings of many previous studies regarding the positive effects of self-management on enterprises; the paper thus argues that the promotional effect of self-management on enterprise management must have an appropriate background and conditions.

### 6.2. Practical Implications

Previous research has failed to answer the question of why some companies can transform themselves and survive when faced with a significant crisis, whereas others fall apart. This paper has significant practical value for understanding the ways in which strategic human resource management can help companies survive and grow in a dynamic environment by enhancing organizational resilience when faced with a crisis and uncertainty.

First, enterprises should actively guarantee that their corporate strategies match their human resource management to ensure that human resources can serve as critical capital to help enterprises survive the crisis and achieve their strategic goals smoothly. The novel coronavirus epidemic is a significant test of enterprise human resource management and continuous operation and development. Companies should optimize their corporate strategies and human resource structures continuously as part of their daily operations and should focus on the power of talent. When facing a crisis, companies should be skilled at exploring the potential opportunities associated with the challenges, thereby improving the cohesiveness of employees, taking full advantage of the creativity of employees, and skillfully using the company’s potential resources so that the company can endure the crisis smoothly; accordingly, the company should actively reflect on the problems and loopholes in the company’s operation after the crisis, further adjust the company’s strategic layout, and be fully prepared to deal with possible crises in the future.

Second, the enterprise should focus on improving employees’ self-efficacy and enhancing their work execution and enthusiasm. Employees are the primary capital of an enterprise and represent the only driving force for the creation of value. In an enterprise, human resource management should focus on adopting people-oriented management policies, cultivating employees’ self-efficacy, and allowing employees to realize that the enterprise values them. This paper explores the role of self-efficacy in enhancing organizational resilience from a practical perspective and shows that the enhancement and utilization of the enterprise’s organizational resilience capability ultimately depends on the power of its employees.

Finally, the enterprise should focus on employees’ self-management capabilities and simultaneously enhance its own internal management capabilities. Previous research has illustrated a variety of benefits of employee self-management on corporate performance. However, based on both theoretical extrapolation and practical research, this paper demonstrates that self-management is not beneficial to organizational development under all conditions. Only when employees’ self-goals and organizational goals are aligned do employees exert their utmost efforts to accomplish overall corporate goals. In management practice, managers should focus on employees’ career development plans and intrinsic needs to ensure that the organization’s strategy matches their jobs and to guarantee that their jobs meet their intrinsic needs.

### 6.3. Limitations and Prospects

This study employs a combination of theoretical derivation and empirical research. It achieves some success regarding both the theoretical and practical aspects of organizational resilience research, but it also faces certain limitations. First, this paper uses only the questionnaire method to obtain sample data, i.e., it relies on a single data source. Future research can employ experimental, interview, and other methods combined with a questionnaire to improve data accuracy. Second, the data used in this study were obtained from employees’ self-reports, and no attention was given to temporal changes when the respondents completed the questionnaires. Although this paper examined the possibility of common method bias using Harman’s one-way analysis of variance method, the results of which were within an acceptable range, the effect of common method bias could not be avoided entirely. Future studies can reduce common method bias by obtaining objective data from companies or enhancing the design of the study. Finally, this study explored only the mediating variable of self-efficacy. Future research can explore other mediating variables associated with the relationship between strategic HRM and organizational resilience from other perspectives with the aim of gradually improving the research on the mechanism underlying the effects of strategic HRM and organizational resilience.

## Figures and Tables

**Figure 1 behavsci-12-00508-f001:**
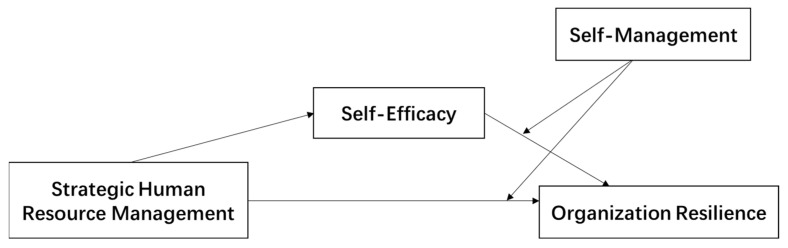
Hypothetical model of the mediating effect of self-efficacy and the moderating effect of self-management.

**Figure 2 behavsci-12-00508-f002:**
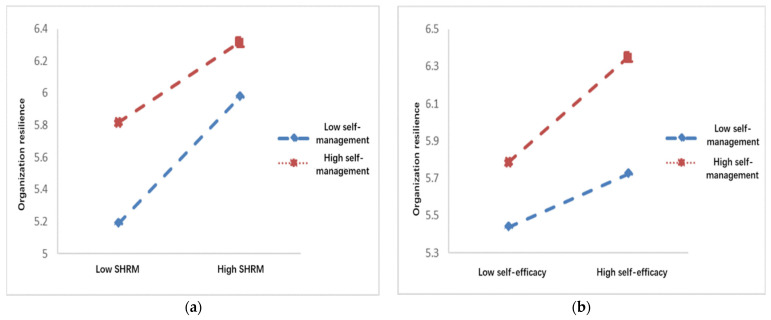
(**a**) The moderating role of self-management in the relationship between strategic human resource management and organizational resilience; (**b**) the moderating role of self-management in the relationship between self-efficacy and organizational resilience.

**Table 1 behavsci-12-00508-t001:** Reliability test results for each variable.

Dimension	*N*	*Cronbach’s α*	*CR*	*AVE*
SHRM	19	0.96	0.96	0.56
SM	10	0.95	0.95	0.66
SE	8	0.96	0.96	0.77
OR	15	0.97	0.87	0.83

Note: SHRM represents Strategic Human Resource Management; SM represents Self-Management; SE represents Self-Efficacy; OR represents Organizational Resilience.

**Table 2 behavsci-12-00508-t002:** Results of validation factor analysis.

Model	X^2^	df	X^2^/df	CFI	IFI	RMSEA	Model Compare	DC2	Ddf
Original Model	3250.633	1223	2.658	0.905	0.905	0.066			
3-factor a	3587.905	1227	2.924	0.889	0.890	0.071	2 vs. 1	337.272 ***	4
3-factors b	4224.300	1227	3.443	0.859	0.860	0.080	3 vs. 1	636.395 ***	4
3-factor c	4346.399	1227	3.615	0.849	0.850	0.083	4 vs. 1	122.099 ***	4
2-factor	4441.853	1228	3.617	0.849	0.850	0.083	5 vs. 1	95.454 ***	3
1-factor	5519.733	1230	4.488	0.798	0.799	0.096	6 vs. 1	1077.880 ***	3

Note: *** denotes *p* < 0.001.

**Table 3 behavsci-12-00508-t003:** Means, variances, and correlation coefficients of the variables.

Means	Standard Deviation	OR	SM	SE	SHRM
5.5402	1.22914	1			
5.6304	0.90327	0.747 **	1		
5.8146	1.07130	0.711 **	0.833 **	1	
5.1990	1.11426	0.722 **	0.640 **	0.676 **	1

Note: ** denotes *p* < 0.01; SHRM represents Strategic Human Resource Management; SM represents Self-Management; SE represents Self-Efficacy; OR represents Organizational Resilience.

**Table 4 behavsci-12-00508-t004:** Mediated model test of self-efficacy.

	OR		OR		SE	
	B	t	B	t	B	t
Gender	0.081	1.030	0.063	0.727	−0.040	−0.484
Age	−0.034	−0.866	−0.001	−0.029	0.070	1.778
Education	−0.031	−0.736	−0.068	−1.546	−0.084	−2.062 *
SHRM	0.450	10.424 ***	0.711	19.952 ***	0.568	17.180 ***
SE	0.459	9.098 ***				
R-sq	0.622		0.5375		0.4751	
F	122.499		108.669		84.628	

Note: * denotes *p* < 0.05; *** denotes *p* < 0.001; SHRM represents Strategic Human Resource Management; SM represents Self-Management; SE represents Self Efficacy; OR represents Organizational Resilience.

**Table 5 behavsci-12-00508-t005:** Decomposition of total utility, direct effects, and mediating effects.

	Effect	BootSE	BootLLCI	BootULCI	Effectiveness Ratio
Indirect effect	0.261	0.042	0.181	0.348	37%
Direct effect	0.450	0.060	0.327	0.562	63%
Total effect	0.711	0.040	0.630	0.787	100%

**Table 6 behavsci-12-00508-t006:** Mediated model tests with moderation.

	OR		SE	
	B	t	B	t
Gender	0.066	0.910	−0.039	−0.484
Age	−0.018	−0.505	0.070	1.779
Education	−0.034	−0.890	−0.084	−2.062 *
SHRM	0.394	9.432 ***	0.567	17.180 ***
SE	0.143	2.020 *		
SM	0.488	6.617 ***		
SHRM * SM	−0.144	−3.130 **		
SE * SM	0.137	3.452 ***		
R-sq	0.684		0.475	
F	99.997		84.627	

Note: * denotes *p* < 0.05; ** denotes *p* < 0.01; *** denotes *p* < 0.001; SHRM represents Strategic Human Resource Management; SM represents Self-Management; SE represents Self-Efficacy; OR represents Organizational Resilience.

**Table 7 behavsci-12-00508-t007:** Direct and mediated effects at different levels of self-management.

	Indicators	Effect	BootSE	BootLLCI	BootULCI
moderating mediating effect	eff1 (M − 1SD)	0.002	0.058	−0.116	0.112
	eff2 (M)	0.082	0.053	−0.019	0.190
	eff3 (M + 1SD)	0.161	0.068	0.037	0.302

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
