# Peer review of "A Study of the Impact of Strategic Human Resource Management on Organizational Resilience"

_behavsci, 2022, doi:10.3390/bs12120508_

Round 1

Reviewer 1 Report

Thank you for the opportunity to review this interesting paper, which I found as well written and addressing an important topic. 

The paper demonstrate an adequate understanding of the relevant literature and cite an appropriate range of literature sources. Also, the paper is built on an appropriate base of theory and  concepts. The research on which the paper is based is well designed and the methods are appropriate. The results are presented clearly and analysed appropriately. Also, the conclusions adequately tie together the other elements of the paper. 

Author Response

Dear Reviewer 1,

Thank you so much  for your kind and positive coments.According to your comments,we still  try  our best to better our manuscript in order to be better qualified for the prestigious journal.Our revisions are shown as the revised edition of our manuscript. 

Reviewer 2 Report

Line 27-28: Statements with a political connotation should not be included in research papers

Line 39: It is not clear the meaning of “domestic literature”. The authors should bear in mind that (if the article would be published), the audience will be from all over the world.

Line 59, 120, 140, so on ...: If the name of the author is included into the text, the within text reference should be placed immediately after.

Line 111-117: This paragraph is from the journal template, and I suppose the authors don’t intend to have it in the paper.

Line 162-202: The paragraph is too long and difficult to follow. Please break it into 2 or 3 shorter ones.

Line 206: At the beginning of the paragraph a capital letter should be placed.

In the methodological section more information should be provided on the sampling, data collection process and data analysis methods.

There are a number of expressions to be revised. Eg:

·       Harman's one-way test

·       verified standard method bias

Table 1: Cronbach Alpha values above 0.9 may point to redundancy. Please comment on this potential issue.

Table 7: Please check the indicator column

It is not clear why references are written in two different styles. Please correct.

Author Response

Dear Reviewer,

Thank you so much for reviewing and commenting our manuscript in such a constructive way.We really appreciate your professioal comments and kind reminders.We pay much importance to your valuable comments and respond to your constructive comments point to point as follows.

Dear Reviewers

Thank you very much for your valuable comments, and we have revised the paper one by one according to your comments.

1.Line 27-28: Statements with a political connotation should not be included in research papers

We have revised the politically oriented sentences in the first paragraph of Introduction,which is shown in line33-36.

2.Line 39: It is not clear the meaning of “domestic literature”. The authors should bear in mind that (if the article would be published), the audience will be from all over the world.

Thank you very much for your kind reminder.We have revised the expressions to the extant literatures in China,wich is shown in line 46 in the second paragraph of the Introduction part.

3.Line 59, 120, 140, so on ...: If the name of the author is included into the text, the within text reference should be placed immediately after.

Thank you so much for your professional commets.All sentences with the author's name are revised and followed by the corresponding references.

4.Line 111-117: This paragraph is from the journal template, and I suppose the authors don’t intend to have it in the paper.

Thank you so much for your such a careful reminder.We removed the irrelevant lines of the Literature review and research hypothesis part.

5.Line 162-202: The paragraph is too long and difficult to follow. Please break it into 2 or 3 shorter ones.

Thanks for your professional and careful comment.According to your instruction,We have divided the third paragraph of 2.1 into four paragraphs as shown in line166-209.

6.Line 206: At the beginning of the paragraph a capital letter should be placed.

Thanks for your careful review and reminder,we have revised the first letter of the first paragraph of 2.2 to be in capital letters as shown in line213.

7.In the methodological section more information should be provided on the sampling, data collection process and data analysis methods.

Thank you so much for your great comment.We have added more information about the sample, data collection and data analysis in red font in sections 3.1, 3.2, 4.1,4.2, 4.4 of the article ( as shown in line393-410,line418-425,line460-468,line497-500).

8.There are a number of expressions to be revised. Eg:

  • Harman's one-way test
  • verified standard method bias

According to your reminder,we have revised the terminologies of the article.

9.Table 1: Cronbach Alpha values above 0.9 may point to redundancy. Please comment on this potential issue.

According to your suggestion,we have added an explanation to 'The Cronbach's α values are above 0.90' in 4.2( shown in line460-468):

Internal consistency tests were conducted based on the criteria of whether the internal consistency coefficient was greater than 0.7 and whether the internal consistency coefficient would increase after the deletion of a question item. There is no redundancy in the terms of each variable。

Cronbach's Alpha if item is deleted:

(C1, .954), (C2, .950), (C3, .951), (C4, .954), (C5, .949), (C6, .947), (C7, .950), (C8, .950), (A1, .970), (A2, .970), (A3, .969), (A4, .969), (A5, .969), (A6, .969), (A7, .969), (A8, .970), (A9, .969), (A10, .968), (A11, .969), (A12, .969), (A13, .969), (A14, .969), (A15, .970), (B1, .945), (B2, .946), (B3, .943), (B4, .944), (B5, .944), (B6, .946), (B7, .946), (B8, .944), (B9, .944), (B10, .952), (F1, .958), (F2, .958), (F3, .958), (F4, .957), (F5, .959), (F6, .958), (F7, .957), (F8, .958), (F9, .958), (F10, .960), (F11, .960), (F12, .960), (F13, .960), (F14, .959), (F15, .958), (F16, .958), (F17, .957), (F18, .958), (F19, .959)。

Note: A for organizational resilience, B for self-management, F for strategic human resource management, C for self-efficacy

10.Table 7: Please check the indicator column

Thank you so much for your careful and professional review,we have revised the table data in table 7( as shown in line531-532).

11.It is not clear why references are written in two different styles. Please correct.

We really appreciate your great review.We have revised the format of the references.( as shown in line736-844).

Reviewer 3 Report

Dear authors
It was my pleasure to review your manuscript entitled “A Study of the Impact of Strategic Human Resource Management on Organizational Resilience” and advise you to prosper your current research project. In my view, your topic has touched on a critical issue in a fascinating context. However, there are many spaces to be improved in terms of argumentation, theoretical background, research method, and findings. I hope my below comments would help you develop your work into groundbreaking research in your domain.

1. The abstract should indicate the innovation of the work.
2. The positioning of the paper is not entirely clear. It is better to explain the gap in this article further.
The introduction should clearly illustrate (1) what we know (the key theoretical perspectives and empirical findings) and what we do not know (major, unaddressed puzzle, controversy, or paradox does the study address, or why it needs to be addressed and why this matters) and (2) what we will learn from the study, and how the study fundamentally changes, challenges, or advances scholars’ understanding. Much sharper problematization is required so that the introduction draws the reader into the paper. The introduction, therefore, needs to do a better job of setting the stage for the articulation of the theoretical contributions of the study. At the end of the introduction, we should have a clear idea of what the paper is about (i.e., its motivation, the gap in understanding that the paper is trying to address, and a summary of theoretical contributions).
Please provide more information on the context of your study. I would like to better understand the context of your empirical work.
3. Theoretical literature has not been considered and reviewed. It is better to observe the connection between the contents. Try to explain everything except the topics in order to establish the necessary coherence.
4. What were the reasons for using this software?
How is the model fitted?
How the validity and reliability of the questionnaire were measured before sending the questionnaires?
What were the reasons for choosing this statistical population?

5. In the discussion section, for each case study that you have identified separately, the results should be written, what effects it has on the main result?

6. The conclusion shows the final results of your research (you need a conclusion for your research). This section is weak.
Please clarify what are the theoretical and practical contributions of your research.
What are the results of your research, and how can it help your community?
Another round of spellchecking by a native speaker is highly recommended.

Reference.
- Using the following references could be beneficial as these add more evidence to the literature review section:

 The Effect of Knowledge Management on the Sustainability of Technology-Driven Businesses in Emerging Markets: The Mediating Role of Social Media. Sustainability, 14(14), 8602.

 Resilience and Knowledge-Based Firms’ Performance: The Mediating Role of Entrepreneurial Thinking. Journal of Entrepreneurship and Business Resilience, 4(2), 7-29.

Best of luck with the further development of the paper.

Author Response

Dear reviewer,

We really appreciate your professional comments and kind reminders.We learn much from you.We cherish your comments and make our best to revise and better our manuscript point to point according to your comments.Our responsed are shown as follows.

1.The abstract should indicate the innovation of the work.

Thanks for your professional reminder.We have revised the abstract and added the innovation points at the end of the abstract as shown in line42-44.

2.The positioning of the paper is not entirely clear. It is better to explain the gap in this article further.The introduction should clearly illustrate (1) what we know (the key theoretical perspectives and empirical findings) and what we do not know (major, unaddressed puzzle, controversy, or paradox does the study address, or why it needs to be addressed and why this matters) and (2) what we will learn from the study, and how the study fundamentally changes, challenges, or advances scholars’ understanding. Much sharper problematization is required so that the introduction draws the reader into the paper. The introduction, therefore, needs to do a better job of setting the stage for the articulation of the theoretical contributions of the study. At the end of the introduction, we should have a clear idea of what the paper is about (i.e., its motivation, the gap in understanding that the paper is trying to address, and a summary of theoretical contributions).

Please provide more information on the context of your study. I would like to better understand the context of your empirical work.

We really appreciate your professional academic instructions.Your suggestions are really constructive in helping us to revise our manuscript. We have extensively revised the Introduction section in red font. It has presented clearly the background, the current status , and the motivation and contribution of the study as shown in line29-69,line89-93,line103-120.

3.Theoretical literature has not been considered and reviewed. It is better to observe the connection between the contents. Try to explain everything except the topics in order to establish the necessary coherence.

Thanks for your patient and professional comments.Aided by your instructions,we have made extensive and full revisions to the introduction part in red to make the connections and logic between the variables clearer as shown in line29-69,line89-93,line103-120.

4.(1)What were the reasons for using this software?

(2)How is the model fitted?

(3)How the validity and reliability of the questionnaire were measured before sending the questionnaires?

(4)What were the reasons for choosing this statistical population?

We really appreciate your logical comments.We have absorbed fully your comments and added much more information about the sample, data collection and data analysis in red font in sections 3.1, 3.2, 4.2, 4.4 .Some supplemented lines are shown as follows.

(1)SPSS and AMOS data analysis software are simple and easy to use, and are widely used in empirical research in management.

(2) According to the results of data analysis in 4.2, the model has good reliability and validity (structural validity, convergent validity, and discriminant validity), and thus the model fit is good. In this paper, the moderated mediation model provided by Winn is used to test the moderated mediation effect and the data are analyzed by using process in spss as a tool in 4.4.(line460-469,497-500)

(3) (a) This paper uses a well-established scale with good reliability and validity that has been validated many times in China, and conducts a small-scale pre-study in 3.2.

(b) We collected 73 samples for a pre-study before the formal research, and removed redundancy and low fit based on the reliability, validity, and cultural context of the pre-study results in 3.1.(line393-410,418-425)

(4) We elaborate on this issue in more detail in red in section 3.1 of the text. Because the purpose of our study is to find organizational resilience improvement paths with broad applicability for Chinese companies, the questionnaire was widely distributed throughout China. (line393-410).

5.In the discussion section, for each case study that you have identified separately, the results should be written, what effects it has on the main result?

Thanks for your constructive and instrumental comments.We have restructured the Discussion part responding to the assumptions in the above text and refined the results for each part to mark them in red in line539-625.

6.The conclusion shows the final results of your research (you need a conclusion for your research). This section is weak.Please clarify what are the theoretical and practical contributions of your research.

What are the results of your research, and how can it help your community?

Another round of spellchecking by a native speaker is highly recommended.

Thanks for your professional comments.We have presented the theoretical contributions and implications correspondingly  in Conclusion 6.1,6.2,shown in 

line 627-635.

7.Reference.

- Using the following references could be beneficial as these add more evidence to the literature review section:

 The Effect of Knowledge Management on the Sustainability of Technology-Driven Businesses in Emerging Markets: The Mediating Role of Social Media. Sustainability, 14(14), 8602.

 Resilience and Knowledge-Based Firms’ Performance: The Mediating Role of Entrepreneurial Thinking. Journal of Entrepreneurship and Business Resilience, 4(2), 7-29.

Thanks for your careful reminders.We have inserted these two references in the text and they are [2], [6],shown in line738-739,748-749.

Round 2

Reviewer 2 Report

.

Reviewer 3 Report

Dear authors

Hope you are doing well. According to the review of this article, the corrections have been made.

Good luck